# Monoallelic variants resulting in substitutions of MAB21L1 Arg51 Cause Aniridia and microphthalmia

Hildegard Nikki Hall[1][☉], Hemant Bengani[1][☉], Robert B. Hufnagel[2], Giuseppe Damante[3], Morad Ansari[4], Joseph A. Marsh[1], Graeme R. Grimes[1], Alex von Kriegsheim[1], David Moore[4], Lisa McKie[1], Jamalia Rahmat[5], Catia Mio[3], Moira Blyth[6], Wee Teik Keng[7], Lily Islam[8], Meriel McEntargart[9], Marcel M. Mannens[10], Veronica Van Heyningen[1], Joe Rainger[11][‡], Brian P. Brooks[2][‡], David R. FitzPatrick[1]*

1 MRC Human Genetics Unit, MRC Institute of Genetics and Molecular Medicine, University of Edinburgh, Edinburgh, United Kingdom, 2 National Eye Institute, National Institutes of Health, Bethesda, MD, United States of America, 3 Department of Medicine, University of Udine, Udine, Italy, 4 South East Scotland Genetic Service, Western General Hospital, Edinburgh, United Kingdom, 5 Ophthalmology Department, Hospital Kuala Lumpur, Kuala Lumpur, Malaysia, 6 University of Leeds, St. James's University Hospital, Leeds, United Kingdom, 7 Department of Genetics, Kuala Lumpur Hospital, Kuala Lumpur, Malaysia, 8 West Midlands Regional Genetics Service, Birmingham Women's and Children's NHS Foundation Trust, Birmingham, England, 9 Medical Genetics, St George's University Hospitals NHS Foundation Trust, London, United Kingdom, 10 Genome Diagnostics laboratory, Department of Clinical Genetics, Amsterdam University Medical Center, Amsterdam, The Netherlands, 11 Roslin Institute, University of Edinburgh, Edinburgh, United Kingdom

☉ These authors contributed equally to this work.
‡ These authors are joint senior authors on this work.
* david.fitzpatrick@ed.ac.uk

**Data Availability Statement:** The RNA sequencing data is available under the GSE166078 series at the NCBI Gene Expression Omnibus (https://www.ncbi.nlm.nih.gov/geo/).

## Abstract

Classical aniridia is a congenital and progressive panocular disorder almost exclusively caused by heterozygous loss-of-function variants at the *PAX6* locus. We report nine individuals from five families with severe aniridia and/or microphthalmia (with no detectable *PAX6* mutation) with ultrarare monoallelic missense variants altering the Arg51 codon of *MAB21L1*. These mutations occurred *de novo* in 3/5 families, with the remaining families being compatible with autosomal dominant inheritance. Mice engineered to carry the p. Arg51Leu change showed a highly-penetrant optic disc anomaly in heterozygous animals with severe microphthalmia in homozygotes. Substitutions of the same codon (Arg51) in *MAB21L2*, a close homolog of *MAB21L1*, cause severe ocular and skeletal malformations in humans and mice. The predicted nucleotidyltransferase function of MAB21L1 could not be demonstrated using purified protein with a variety of nucleotide substrates and oligonucleotide activators. Induced expression of GFP-tagged wildtype and mutant MAB21L1 in human cells caused only modest transcriptional changes. Mass spectrometry of immunoprecipitated protein revealed that both mutant and wildtype MAB21L1 associate with transcription factors that are known regulators of PAX6 (MEIS1, MEIS2 and PBX1) and with poly(A) RNA binding proteins. Arg51 substitutions reduce the association of wild-type MAB21L1 with TBL1XR1, a component of the NCoR complex. We found limited evidence for mutation-specific interactions with MSI2/Musashi-2, an RNA-binding proteins with

**Funding:** H.N.H. is funded by a Wellcome Trust fellowship (205171_Z_16_Z). D.R.F. is supported by Medical Research Council (MRC) University Unit program grant awarded to the University of Edinburgh. Funding for UK10K was provided by Wellcome under award WT091310.

**Competing interests:** The authors declare no conflicts of interest.

effects on many different developmental pathways. Given that biallelic loss-of-function variants in *MAB21L1* result in a milder eye phenotype we suggest that Arg51-altering monoallelic variants most plausibly perturb eye development via a gain-of-function mechanism.

## Introduction

The gene *mab-21* was identified through its ability to rescue the *Caenorhabditis elegans* male abnormal 21 mutants, characterised by a homeotic transformation of the male-specific peripheral sense organs [1]. 11 human paralogs of mab-21 have been identified each with a nucleotidyltransferase domain [2]. The best studied, *CGAS* [MIM 613973], functions in the innate immune system as a sensor of aberrant cytosolic DNA. The binding of short dsDNA induces a conformation change that activates enzymatic production of a cyclic dinucleotide which then functions as a second messenger in the interferon response cascade [3].

The mab-21 paralog, *MAB21L1* [MIM 601280], is a single exon gene located in an intron of *NBEA* [MIM 604889] which is transcribed on the opposite strand. Biallelic loss-of-function mutations in *MAB21L1* cause a developmental disorder characterized by corneal dystrophy, microcephaly, cerebellar hypoplasia and genital anomalies [MIM 618479] [4,5]. The carrier parents of affected individuals were reported to be normal. *Mab21l1* null mice are viable but show severe bilateral microphthalmia with a small malformed lens and absence of the iris and ciliary body [PMID 12642482]. Null mice also show delayed calvarial development and male infertility with hypoplasia of the preputial glands [6,7]. Heterozygous mice apparently normal. Homozygosity for an early frameshift mutation in zebrafish *mab21l1* resulted in a late embryonic degeneration of the cornea and subsequently the lens [PMID 33570754]. The crystal structure of MAB21L1 indicates a cGAS-like capacity for catalytic activation via ligand binding although both the oligonucleotide activator and the nucleotide product are currently unknown [8].

We and others have previously reported heterozygous *de novo* missense mutations in *MAB21L2* [MIM 604357], the closest human homolog of *MAB21L1*, associated with severe bilateral eye malformations and skeletal anomalies [MIM 615877] [9–11]. These variants altered Arg51 with the most severe phenotype associated with Arg51Cys substitutions. A mouse model of this genotype resulted in a phenotype that recapitulated the human disease [12]. *Mab21l2* null mice have severe eye malformations and body wall defects with heterozygous null mice being normal [13].

Here we report monoallelic missense variants that are absent for gnomAD and result in substitution of Arg51 or, in a single case, Phe52 residues of MAB21L1 in families with severe aniridia [MIM 106210], a phenotype associated with monoallelic mutations in *PAX6* [MIM 607108], and/or microphthalmia. An apparently unrelated family has been recently reported with a heterozygous missense variant in *MAB21L1* identical to one that we have identified (c.152G>T p. (Arg51Leu)) associated with microphthalmia and aniridia [14] which provides strong support for the genotype-phenotype association. We present a mouse model of one of these mutations and study the effect of the mutant proteins on the transcriptome and protein interactome using inducible expression of tagged protein in human cells. The results are most consistent with a gain-of-function effect in Arg51-substituted MAB21L1 during embryogenesis.

## Materials and methods

### Recruitment, consent and mutation analysis

This project used clinical information and biological samples from individuals referred to the Medical Research Council (MRC) Human Genetics Unit Eye Malformation Study. Informed

written consent for research was obtained from all families. This cohort was collected and maintained using protocols approved by the Scotland A UK Multicentre Research Ethics Committee, references 06/MRE00/76 and 16/SS/0201. The causative variants were identified using a combination of sequencing approaches: whole exome analysis and candidate gene panel sequencing in the Wellcome Sanger Institute as part of the rare disease component of the UK10K project as described [15] and Sanger sequencing (for details see **Results** and **S1 Table in S7 File**). Samples from two families were referred following discussions with the corresponding author for clinical testing in the NHS South East Scotland Regional Genetics Services using MiSeq sequencing of a targeted gene panel which included *MAB21L1*. All variants were validated using Sanger sequencing of PCR products amplified directly from genomic DNA and were nomenclature-confirmed (https://variantvalidator.org/) (**S2 Table in S7 File**). All variant numbering is based on the human reference sequences GRCh38 NC_000013.11 (genomic, chr13). For each of the missense variants SIFT [16], PolyPhen [17], CADD [18] and REVEL [19] scores were generated using the DECIPHER web tool [20].

## Structural analysis of mutations

The effects of missense mutations were modelled using the crystal structure of MAB21L1 (PDB ID: 5EOM) using FoldX 5.0 [21], which was recently shown to be the top-performing method for the identification of pathogenic missense mutations that affect protein stability [22], using all default parameters and averaging over 10 replicates.

## Cloning, protein purification and enzymatic assay

Wild-type human MAB21L1 and the substitution p.(Arg51Leu) were amplified from control and patient DNA respectively and cloned in frame into the pGEX 6P1 vector (GE Life-Sciences). Purified protein was isolated from induced *E. coli* strain BL21 cultures as outlined in **Supplemental Materials and Methods.** Human OAS1 protein was used as a positive control in the enzymatic assay. A colorimetric method was used to quantitate the amount of pyrophosphate (PPi) product released upon completion of the enzymatic reaction as described [23] and detailed in **Supplemental Materials and Methods**. The resulting chromophore molybdenum blue produced was quantified by spectrophotometry at A580 nm.

## Generation and RNA-based analysis of inducible human cell lines

Full-length human MAB21L1 and Arg51Leu and Arg51Gln substituted forms were amplified from the control and patient DNA and cloned downstream of green fluorescence protein (GFP) in the Gateway pcDNA-DEST53 vector according to the manufacturer's protocol, resulting in an N-terminal fusion protein. Stable cell lines were generated, selected and maintained using Human Embryonic Kidney (HEK)-293 cells with the Flp-In T-REx system (ThermoFisher) according to the manufacturer's guidelines. Details of the subcellular fractionation and Western blotting procedures are provided in **Supplemental Materials and Methods**. RNA sequencing used total RNA extracted from two biological replicates of each cell line after 12 hrs of 1 μg/ml tetracycline treatment using the RNeasy kit (QIAGEN). Random primed cDNA from poly(A) selected RNA was converted into an Illumina sequencing library using RNA Library Prep Kit from Illumina (E7420, NEB, USA) in conjunction with NEBNext® Multiplex Oligos for Illumina (E7335/E7500, NEB, USA). and single-end 50-base pair (bp) reads were generated using a NextSeq 500 (Illumina Inc, SY-415-1002). Transcript-level quantitation was performed using Salmon (v0.8.2) against the GRCh38 Ensembl reference transcriptome (release-89). Transcript-level counts were summarized to gene level using the

Bioconductor package tximport (v1.4.0). Differential expression analysis was performed with the Bioconductor package DESeq2 (v1.30.0) using the Wald significance tests.

## Immunoprecipitation-mass spectrometry

Three biological replicates of HEK-293-Flp-In T-Rex cells tagged with EGFP, EGFP-MAB21L1 or mutant EGFP-MAB21L1 were seeded in T-75 flask in culturing media supplemented with 1 μg/ml tetracycline. Cells were harvested by trypsin-EDTA, washed by PBS after 12 hrs of tetracycline treatment. Cell lysis and GFP pulldown was perform using GFP Tag Immunomagnetic Beads (Sino Biologicals) according to manufacturer instructions. The pull-down beads were subjected to mass spectrometric analysis and raw data was analysed by the MaxQuant and Andromeda software package as described [24], using the pre-selected conditions for analysis (specific proteases, 2 missed cleavages, 7 amino acids minimum length). Detailed Mass Spectrometry analysis is provided in **Supplemental Materials and Methods.**

## Immunoprecipitation-western blotting

HEK-293-Flp-In T-Rex cells tagged with EGFP, EGFP-MAB21L1 or mutant EGFP-MAB21L1 were seeded in T-25 flask in culturing media supplemented with 1 μg/ml tetracycline. Cells were harvested by trypsin-EDTA, washed by PBS after 12 hrs of tetracycline treatment and lysed with Nonidet P-40 lysis buffer (50mM Tris, pH 8.0, 150mM NaCl, 1.0% Nonidet P-40) in the presence of protease inhibitor(Roche Applied Science) For each immunoprecipitation, 400 μl of cell lysate were incubated with anti-TBL1XR1 antibody (ab24550,Abcam) and anti-MSI2 antibody(ab76148,Abcam) for 5 h at 4˚C. Then 20 μl of Dynabeads protein A (Thermo Fischer) were added and rotated for 2 h at 4˚C. Bound immune complexes were washed three times with phosphate-buffered saline. For immunoprecipitation of GFP-tagged proteins, Cell lysis and GFP pulldown was perform using GFP Tag Immunomagnetic Beads (Sino Biologicals) according to manufacturer instructions. The immune-complexes were analysed by Western blotting.

## Generation and phenotyping of mouse model

All mouse work complied with United Kingdom Home Office regulations, with study protocols approved under Home Office project licences (60/4424, P1914806F). CRISPR-Cas9 gene editing methodology was used to introduce a targeted mutation of the Arg51 residue of *Mab21l1* in C57BL/6JCrl zygotes. The CRIPSR design and breeding strategy for the colony are detailed further in **Supplemental Materials and Methods** and **S1 Fig**. Adult mutant and control mice were examined at 2–3 months of age unless otherwise stated, using; slit lamp bio microscopy, indirect ophthalmoscopy, Icare tonometry (intraocular pressure measurement) and endoscopic fundus imaging, all as described [25]. On fundus images, 2D optic disc size was measured semi-automatically using the Vampire Annotation Tool [26]. Optical coherence tomography (OCT) using Spectralis (Heidelberg Engineering) was performed as described [27]. For histology mice were culled and enucleated eyes were preserved in Davidson's fixative and then wax embedded, sectioned and stained with Heamatoxylin and Eosin as previously described [28].

# Results

## Identification of MAB21L1 monoallelic missense variants altering Arg51

As part of the rare disease component of the UK10K Study [15] 384 mostly unrelated individuals with bilateral eye malformations were batch sequenced using a targeted pull-down of 1000

candidate genes, 100 of which had been chosen on the basis of their involvement in eye development. Filtering for rare variants within these 100 genes identified a heterozygous plausible deleterious variant c.152G>A p.(Arg51Gln) in MAB21L1 (ENST00000379919.6:c.152G>A, ENSP00000369251.4:p.Arg51Gln: Sift; Deleterious (0), PolyPhen; Probably damaging (0.999), CADD 30, REVEL 0.542) in a single individual (Family 511: II:1, **Fig 1A and 1B**) with bilateral profound aniridia and microphthalmia (**Table 1**, **Fig 1C**). In this family the eye malformations were inherited as an autosomal dominant disorder and the *MAB21L1* variant segregated with the phenotype (**Fig 1A and 1C**). The same c.152G>A p.(Arg51Gln) variant was identified in an individual referred from south-east Asian (Family 96571: II:1, Fig 1A) for clinical investigation of bilateral, severe microphthalmia (**Table 1**). This variant was subsequently also identified in an affected brother (**Fig 1A**). The affected offspring had inherited the variant from their unaffected mosaic father (**Fig 1A**, **Table 1**).

Subsequence sequencing of DNA from unrelated affected individuals referred to the MRC Human Genetics Unit Eye Malformations Study identified an individual with sporadic partial aniridia and microphthalmia (Family 1434: II:1, **Fig 1A and 1C**) associated with *de novo* occurrence of *MAB21L1* c.152G>T p.(Arg51Leu) (ENST00000379919.6:c.152G>T, ENSP00000369251.4:p.Arg51Leu: Sift; Deleterious (0), PolyPhen; Probably damaging (0.999), CADD 29.6, REVEL 0.682). The same allele was identified in an individual referred with familial aniridia (Family 3413: I:1, **Fig 1A**) however samples from other affected members of this family were not available for testing. A further *de novo* missense variant c.152G>C p.(Arg51-Pro) (ENST00000379919.6:c.152G>C, ENSP00000369251.4:p.Arg51Pro: Sift; Deleterious (0), PolyPhen; Probably damaging (1), CADD 31, REVEL 0.694) was identified in an individual with a sporadic milder aniridia-spectrum eye malformation (Family 592). Finally, a variant affecting the adjacent codon, c.155T>G p.(Phe52Cys) (ENST00000379919.6:c.155T>G, ENSP00000369251.4:p.Phe52Cys: Sift; Deleterious (0), PolyPhen; Probably damaging (0.969), CADD 32, REVEL 0.745), was identified in an individual with sporadic microphthalmia and aniridia (Family 5531: I:1, **Fig 1A**, **Table 1**). Parental samples were not available for testing in Family 5531. Sequencing chromatograms are provided (**S2 Fig**). None of these variants have been observed in publicly available variant databases.

**Clinical phenotype.** Pedigrees and clinical images are provided (**Fig 1A and 1C**), as well as detailed clinical descriptions (**Supplemental Clinical Descriptions**). The phenotypic features (**Table 1**) are summarised here. All 6 families had aniridia and/or microphthalmia. Aniridia was present in 4/6 families, microphthalmia in 4/6; and both in 3/6.

**Aniridia.** The aniridia was partial (moderate loss of iris tissue) in 2/6, unspecified in 1/6, and profound in 1/6. None of the families had a documented normal iris: 1/6 had a milder iris phenotype consisting of an irregular pupillary margin, and the remaining 1/6 had severe microphthalmia.

**Microphthalmia/MAC spectrum.** Of the MAC spectrum features seen in 4/6: 1/6 had a family member with bilateral severe microphthalmia, no view of the internal ocular structures and no perception of light. 1/6 had a microphthalmia with a chorioretinal colobomata and microcornea (in all 3 family members).

**Other ocular features.** 4/6 had nystagmus, 3/6 with confirmed foveal hypoplasia; in the remaining 2/6 only limited phenotypic data was available. 3/6 families had cataract, 2/6 with lens instability or subluxation. None had aphakia. 1/6 had glaucoma. 2/6 were described as having keratopathy or an opaque cornea, but as both individuals had microphthalmia and one had phthisis this is difficult to interpret. 2/6 had optic disc anomalies, including congenitally excavated and hypoplastic nerves.

**Non-ocular features.** There were no non-ocular phenotypic features of note. In particular, none of the affected individuals were reported to have genital anomalies.

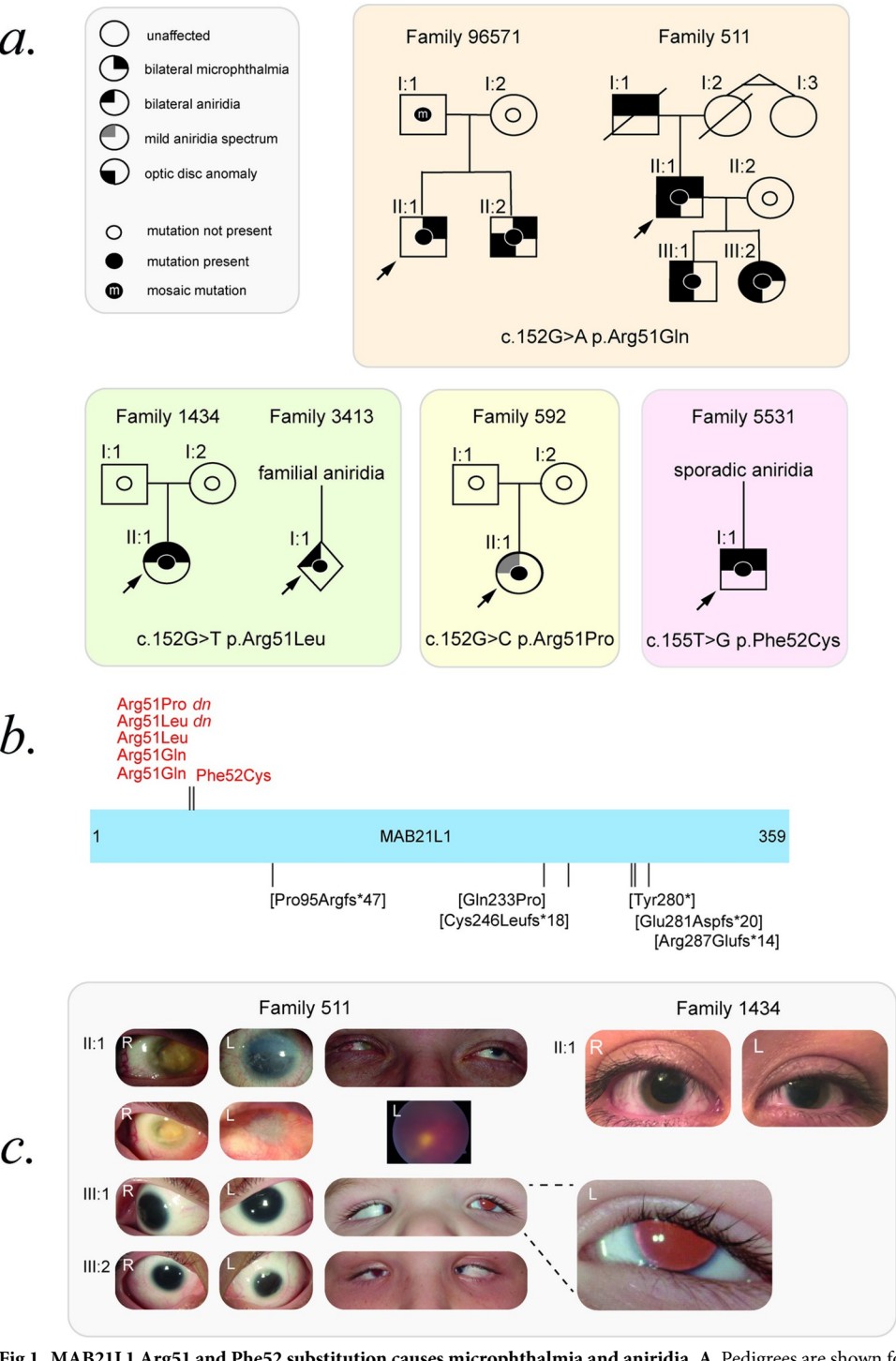

**Fig 1. MAB21L1 Arg51 and Phe52 substitution causes microphthalmia and aniridia. A**. Pedigrees are shown for the six families with MAB21L1 variants and bilateral microphthalmia and/or aniridia. The pedigrees are ordered by variant: c.152G>A p.(Arg51Gln) (orange shaded box), c.152G>T p.(Arg51Leu) (green shaded box), c.152G>C p. (Arg51Pro) (yellow shaded box) and c.155T>G p.(Arg52Cys) (pink shaded box). A key to the pedigree symbols is shown to the left (grey shaded box). **B**. A schematic of MAB21L1 represented as a linear bar and with the first and last amino acid residue numbered. The linear positions of all pathogenic variants are shown: The monoallelic variants in this study are detailed above (red text) and the published biallelic variants are detailed below (bracketed black text). **C**. Clinical images of individuals with MAB21L1 Arg51-related eye malformations. R, right eye; L, left eye. Family 511 all have profound aniridia, microcornea, choroidal coloboma (just visible in II:1's L fundus photo) and optic disc anomalies. The progression of disease in II:1 over one decade is shown between with the upper and lower photos, with

worsening of phthisis in the right and pannus in the left. An enlarged retroilluminated image of III:1's L eye is shown highlighting near-total aniridia. Individual 1434 II:1, showing bilateral partial aniridia and microphthalmia, worse on the L. Abbreviations: dn, *de novo*. Nucleotide and amino acid numbering are based on GenBank NM_005584.5 and GenPept NP_005575.1, respectively.

## Enzymatic function of MAB21L1

Wild-type and mutant (p.Arg51Leu) MAB21L1 was purified from *E. coli* in order to determine whether its predicted enzymatic function could be detected. Using an assay for colorimetric detection of pyrophosphate release [23] we could detect strong activity with 2'-5'-oligoadenylate synthase (OAS) purified by the same methods with ATP as substrate and RNA as activator molecule (**Fig 2A**). However, no MAB21L1-associated nucleotidyltransferase activity was detected using various nucleoside triphosphates as a substrate along with DNA or RNA as activator molecules (**Fig 2A**).

## Structural analysis of MAB21L1 residue substitutions

We analysed the protein structural context of MAB21L1 substitutions reported above and the single reported biallelic missense variant [4] (**Fig 3B**) incorporating previously reported MAB21L2 residue substitutions associated with monoallelic or biallelic genotypes [10] (**S3 Fig**). There is a clear clustering of heterozygous variants, centred at Arg51. All the mutations are predicted to be destabilizing to protein structure (**S4 Table in S7 File**), except the recessive MAB21L2 substitution Arg247Gln; however, previous experimental work has conclusively demonstrated the destabilizing nature of this variant [8]. The pathogenicity of the recessive variants can almost certainly be explained by a simple loss of function caused by protein destabilization. However, while the heterozygous variants are all predicted to be somewhat disruptive, their clustering suggests a specific effect that involves this region. Thus, it seems plausible that the Arg51 substitutions are altering an interaction with another protein. It is also interesting that Arg51Pro has the mildest protein structural effect and appears to cause a milder phenotype than the other *MAB21L1* heterozygous missense variants (**Fig 1A, Table 1**). Moreover, FoldX predicts a strong destabilizing effect of the Phe52Cys substitution, thus further indicating the structural relevance of this region.

## Creation and Analysis of stable cell lines with inducible expression of wild-type and mutant MAB21L1

We created multiple independent tetracycline-inducible cell lines expressing wild-type MAB21L1 and Arg51leu and Arg51Gln variants as full-length GFP-tagged fusion proteins. Analysis of nuclear and cytoplasmic fractions revealed that MAB21L1 was present in both fraction with no evidence of mislocalisation of mutant forms (**Fig 2B**). RNAseq was used to assess the effect of *MAB21L1* mutant variants on gene expression. Relatively few genes showed consistent differences between mutant and wild-type *MAB21L1* (**Fig 2C**). Of these only *SPARC* (secreted protein acidic and rich in cysteine [MIM 182120]) had any link to eye disease [29,30]. SPARC was significantly upregulated in mutant cell lines compared to wild-type. SPARC has been reported as a PAX6 interactor [31] so we used transient transfection to over-express exogenous *PAX6* in the wild-type and mutant MAB21L1 cells to see if this had any effect on the mutant-specific *SPARC* upregulation. Interestingly PAX6 had no effect on *SPARC* expression in mutant cells but induced significant downregulation in cells expressing wild-type GFP-MAB21L1 (**Fig 2D**). This would be consistent with wild-type MAB21L1 having a role in PAX6 mediated repression of SPARC and that function being lost with Arg51 substitution.

**Table 1. Clinical and molecular features of individuals with *MAB21L1* heterozygous variants.**

| Family ID | 96571 | | 511 | | | 1434 | 3413 | 592 | 5531 |
|---|---|---|---|---|---|---|---|---|---|
| Case | II:1 | II:2 | II:1 | III:1 | III:2 | II:1 | I:1 | II:1 | I:1 |
| Sex | male | male | male | male | female | female | ND | female | male |
| GRCh38: NC_000013.11 | g.35475987C>T | | g.35475987C>T | | | g.35475987C>A | g.35475987C>A | g.35475987C>G | g.35475984A>C |
| GenBank: NM_005584.5 | c.152G>A | | c.152G>A | | | c.152G>T | c.152G>T | c.152G>C | c.155T>G |
| GenPept: NP_005575.1 | p.(Arg51Gln) | | p.(Arg51Gln) | | | p.(Arg51Leu) | p.(Arg51Leu) | p.(Arg51Pro) | p.(Phe52Cys) |
| Inheritance | paternal[a] | paternal[a] | ND[b] | paternal | paternal | de novo | ND | de novo | ND[c] |
| Growth | | | | | | | | | |
| Birth Weight z score | 2.9 | 3.05 | | | | | | | |
| Age at last assessment, years | 7.5 | 2 | 48 | 13 | 11 | 23 | ND | 11 | ND |
| Height z score | 2.08 | 1.2 | | | | | | | |
| Weight z score | 1.31 | 0.76 | | | | | | | |
| OFC, cm | 51 | 49.5 | | | | | | | |
| Ocular features | | | | | | | | | |
| Microphthalmia | BL severe | BL | BL | | BL | BL | | | BL |
| Coloboma | | | LE small, inferior to disc | RE infero-temporal choroidal; LE small, temporal to disc | LE nasal choroidal | | | | |
| Aniridia | | | BL | BL | BL | BL | BL | | BL |
| Irregular pupil margin | | | | | | | | BL | |
| Microcornea, mm | | | BL | BL: 6 | BL: 6 | | | | |
| Keratopathy | LE opaque vascularised cornea | RE opaque cornea | BL progressive | | | | | | |
| Glaucoma | | | | | | BL with surgery LE | | | |
| Cataract | no view | | LE previous lensectomy | | BL with RE lenticonus, LE lens instability | BL with progressive subluxation | | BL | |
| Nystagmus | | LE | BL | BL | BL | BL | | BL | |
| Foveal hypoplasia | | LE | insufficient view | | | BL | | BL | |
| Optic disc anomaly | | LE | RE no view; LE congenitally excavated appearance, normal colour | BL gray hypoplastic | BL gray hypoplastic | | | | |
| Myopia | | | | | | BL high | | BL | |

*(Continued)*

**Table 1.** (Continued)

| Family ID | 96571 | | 511 | | | 1434 | 3413 | 592 | 5531 |
|---|---|---|---|---|---|---|---|---|---|
| Case | II:1 | II:2 | II:1 | III:1 | III:2 | II:1 | I:1 | II:1 | I:1 |
| Sex | male | male | male | male | female | female | ND | female | male |
| Additional details | BL no view of internal structures | RE no view of internal structures | RE phthisis of unknown cause | | | | limited phenotypic data | | limited phenotypic data |
| Visual acuity | BL NPL | BL PL | | | | BL HM | | | |
| Non-ocular features | | | | | | | | | |
| | hyperthyroidism (also in mother); normal MRI brain scan, echocardiogram and renal ultrasound | | | | | BL mild sensorineural hearing loss, normal MRI brain scan | | | |

[a], the unaffected father was gonosomal mosaic (at a level of approximately 27 percent) for the variant.

[b], his deceased father had microphthalmia and aniridia.

[c], sporadic case of aniridia.

Abbreviations are BL, bilateral; HM, hand movements; LE, left eye; MRI, magnetic resonance imaging; ND, not determined; NPL, no perception of light; PL, perception of light; RE, right eye.

### Detection of wild-type and mutant-specific protein interactions

To identify MAB21L1 interactors that are specific either to wild-type or mutant protein we performed immunoprecipitation followed by mass spectrometry (IP-MS) on biological triplicates derived from independent clones of the inducible cell lines expressing GFP alone, wild-type MAB21L1, Arg51Leu MAB21L1 and Arg51Gln MAB21L1. More than 1000 proteins were identified using IP-MS but most were non-specific or inconsistently associated with the genotypes (**Fig 3A**). The levels of MAB21L1 were similar in wild-type and mutant pull-down samples suggesting uniform pulldown (**Fig 3B**). 72 proteins showed consistent association with all forms of MAB21L1 with no association with GFP alone. Pathway analysis of these proteins revealed a significant over-representation of RNA-binding proteins and TALE-like homeodomain containing proteins (**Table 2**). Indeed, four of the five most abundant proteins were transcription factors of this latter class (MEIS1, MEIS2, PBX1 and PBX3) (**Fig 3D**). A component of the NCor co-repressor complex, TBL1XR1, was the only wild-type MAB21L1-specific protein identified (**Fig 3E**). Western blot of the GFP IP using an antibody raised against TBL1XR1 showed differential, but not exclusive, binding to wild-type MAB21L1 (**Fig 3F**). Reciprocal immunoprecipitation using anti- TBL1XR1 antibody was not able to detect either mutant or wild-type forms of MAB21L1 using western blot (**S4 Fig**). MSI2/Musashi-2 was one of only three proteins showing apparently exclusively association with the mutant forms of MAB21L1 (**Fig 3G,** the other proteins being LRRFIP1 and GALNT2, **S5 Fig**). Although we were unable to confirm this interaction using reciprocal IP with a MSI2 antibody on western blot (**Fig 3H**), the reciprocal IP-MS using this antibody identified MAB21L1 derived peptides in each replicate of the mutant forms of MAB21L1 but only one of the wild-type replicates. This can be considered only limited evidence of a gain of function interactions since the GFP-only MS-IP showed peptides in two of the replicates, presumably derived from endogenous MAB21L1 in HEK293 cells.

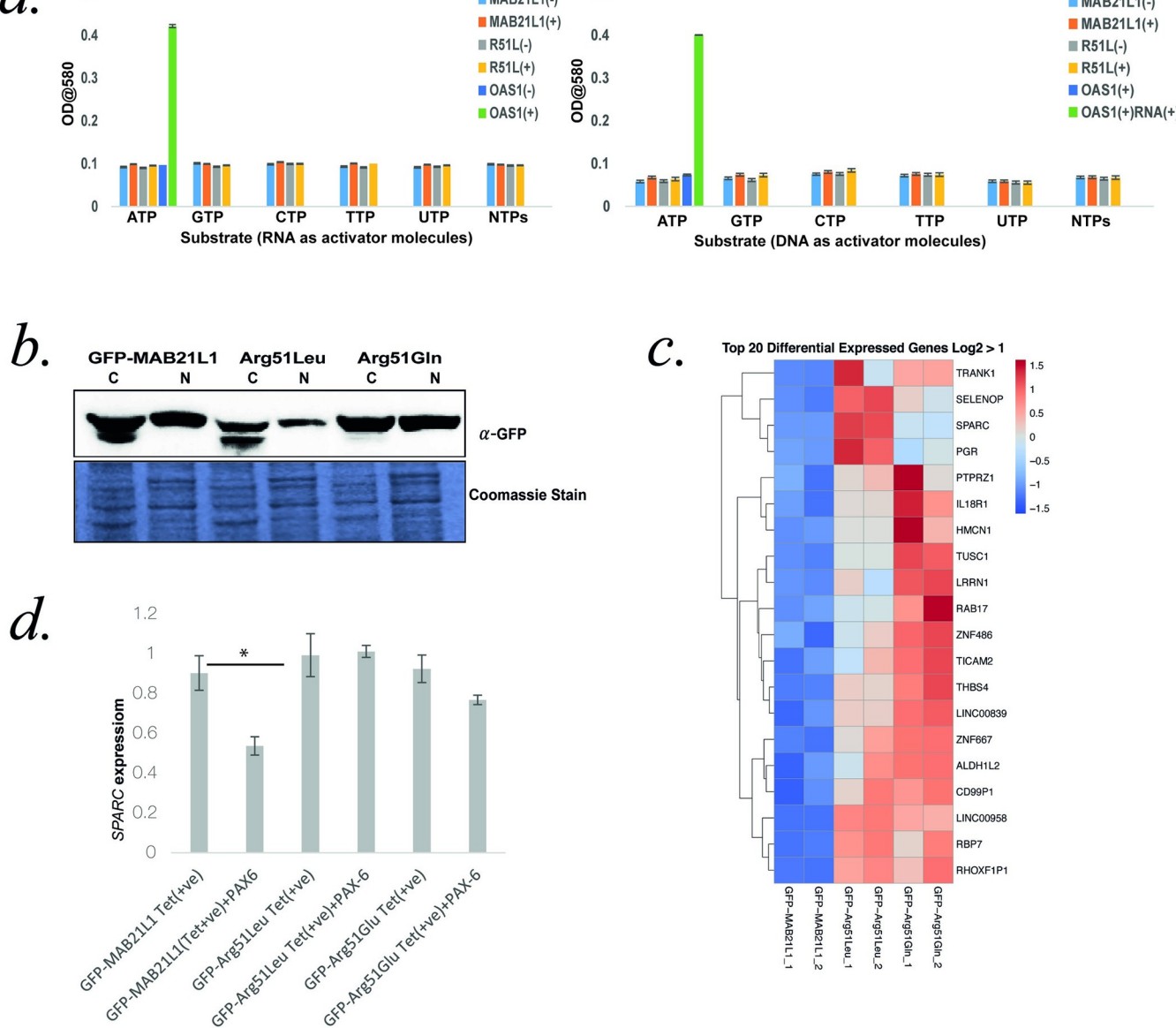

**Fig 2. A. Nucleotidyltransferase activity:** Graph showing the absence of nucleotidyltransferase activity in MAB21L1 and its mutant form Arg51Leu purified protein. OAS1 protein purified in the same way is a positive control and when incubated with ATP and double-stranded RNA (dsRNA), significant pyrophosphate release is detected indicating nucleotidyl transferase activity. MAB21L1 and Arg51Leu showed no activity with either ATP, CTP, GTP, UTP used as substrate separately or as an equal mixture of NTPs using DNA or RNA as an activator. The error bars represent standard errors.**B: Cellular fractionation:** Western blot analysis of cytoplasmic (C) and nuclear(N) extracts from HEK293-Flp-In cells with Tetracyclin (TET) inducible expression of GFP-tagged wild-type and mutant MAB21L1(Arg51Leu and Arg51Gln).Wild type and mutant proteins were present in cytoplasm(C) as well as nuclear fraction(N) as detected by anti-GFP antibody. Representative Coomassie stain gel image is shown.**C: Differential Gene Expression:** Gene expression analysis by RNA Sequencing performed on GFP-tagged wild-type and mutant MAB21L1 (Arg51Leu and Arg51Gln) cells. Heatmap showing top 20 differentially expressed genes in the datasets (padj < .05 and Log2F>1). The RNA sequencing data is available under the GSE166078 series at the NCBI Gene Expression Omnibus (https://www.ncbi.nlm.nih.gov/geo/). **D: Effect of PAX6 overexpression on SPARC transcripts levels:** SPARC transcripts levels were quantified using quantitative RT-PCR using cells expressing GFP tagged Wild type and mutant MAB21L1 with or without overexpressing PAX6. GAPDH transcripts levels were used as normalization control. The levels of SPARC transcripts were significantly reduced in GFP tagged Wild type MAB21L1 cells in presence of overexpressed PAX6. There was no significant difference in the mutant cells in presence or absence of overexpressed PAX6.

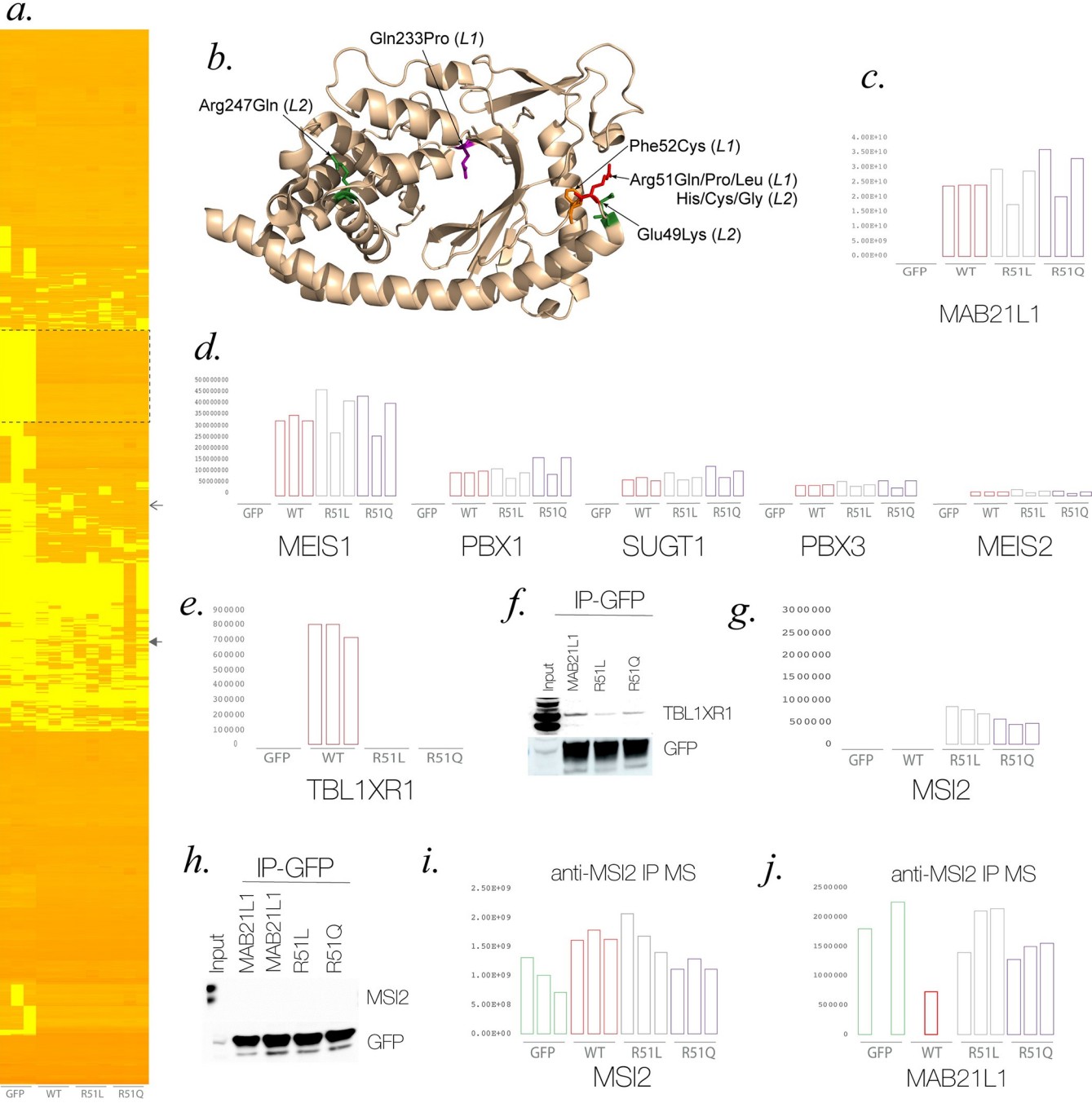

**Fig 3. The Effect of disease associated missense variants on protein structure and interactions. A.** A heatmap of the log-transformed quantitative mass spectroscopy (MS) results of biological triplicates of anti-GFP immunoprecipitates (IP) of control (GFP), tagged wild-type (WT) and mutant MAB21L1 (R51L and R51Q) from HEK293 cells. **B.** Representation of the structure of MAB21L1 with the position of the amino acid substitutions seen in *MAB21L1* (L1) and *MAB21L2* (L2) annotated. **C-E** Graphs showing the levels of the following classes of proteins in the GFP IP-MS from biological triplicates: **C.** MAB21L1. **D.** The five most abundant proteins interacting with WT, R51L and R51Q forms of MAB21L1 (MEIS1, PBX1, SUGT1, PBX3 & MEIS2), the dotted line box indicates the position of this class of protein on the heatmap. **E.** Wild-type specific interactor (TBL1XR1) and other components of the NCor complex (NCOR & HDAC3), the position in the heatmap is indicated by the closed arrowhead. **F.** Western blot analysis of the anti-GFP IP using the anti-TBL1XR1 antibody showing differential but not exclusive binding of the wild-type MAB21L1 compared to the mutant forms **G.** Mutant specific interactor MSI2/ Musashi-2), the position in the heatmap is indicated by the open arrowhead. **H.** Western blot analysis of the anti-GFP IP using the anti-MSI2 antibody was unable to detect interaction with wild-type or mutant forms of MAB21L1 **I-J.** anti-MSI2 IP-MS analysis **I.** MSI2-derived peptides were present in all replicates and cell-lines in the anti-MSI2 IP **J.** MAB21L1-derived peptides were detectable in all replicates of the mutant forms of MAB21L1 but in only one replicate for the wild-type. Surprisingly peptides derived from endogenous MAB21L1 were detectable in two of the three GFP-only biological replicates.

**Table 2. Significantly enriched terms relating to MAB21L1-interacting proteins using DAVID Functional Annotation Chart.**

| Category | Term | Genes | Fold Enrich | Bonferroni |
|---|---|---|---|---|
| GO:0044822 | poly(A) RNA binding | RBM26, NCBP1, POP1, SPATS2, MRPS21, ERAL1, MRPL37, NPM3, DIAPH1, SARS2, ZC3H7A, RBMX2, NUSAP1, FASTKD5, FLNB, METTL16, SNTB2 | 3.68 | 1.07E-03 |
| UP_KEYWORDS | Phosphoprotein | RBM26, PDXDC1, POP1, FLII, GPS1, RNF219, STK4, PFAS, SMG5, TOR1AIP1, STK3, DNAJB1, ATXN3, IPO8, ZC3H7A, SALL2, RBMX2, PHKG2, NUSAP1, FLNB, METTL16, SKP2, MARK3, CEP55, EIF2A, JAGN1, NCBP1, HMGCS1, RIOK3, ZNF281, GLMN, SPATS2, RECQL, PYCR2, CDC7, ERAL1, HAUS5, NPM3, GTF2F2, GPN1, DIAPH1, FASTKD5, RLIM, UBA2, CDC42EP1, GARS, SUGT1, SNTB2 | 1.66 | 1.12E-03 |
| UP_KEYWORDS | Acetylation | RBM26, FLII, STK4, SMG5, STK3, NUSAP1, FLNB, SKP2, EIF2A, NCBP1, HMGCS1, GLMN, RECQL, PYCR2, HAUS5, NPM3, GTF2F2, GPN1, DIAPH1, PRPF4, PPP5C, SARS2, GGCX, FASTKD5, RLIM, UBA2, GARS, SUGT1 | 2.34 | 1.89E-03 |
| UP_SEQ_FEATURE | DNA-binding region: Homeobox; TALE-type | MEIS1, PBX3, MEIS2, PBX1 | 56.51 | 1.25E-02 |
| UP_KEYWORDS | Nucleus | POP1, FLII, GPS1, MRFAP1, STK4, SMG5, TOR1AIP1, STK3, DNAJB1, ATXN3, IPO8, ZC3H7A, SALL2, NUSAP1, SKP2, ZSCAN18, ZNF460, NCBP1, ZNF281, RECQL, PBX3, CDC7, NPM3, GTF2F2, MEIS2, GPN1, PBX1, PRPF4, PPP5C, MEIS1, RLIM, UBA2, SUGT1, INTS9 | 1.85 | 0.015 |

Fold Enrich, fold enrichment over homo sapiens background list; Bonferroni, p value corrected for multiple testing using the Bonferroni method.

## Generation and phenotyping of mice with *Mab21l1* p.Arg51Leu substitution

We used zygotic genome editing to create a mouse line harbouring *Mab21l1* p.Arg51Leu substitution (*Mab21l1$^{R51L/+}$*) (**S1 Fig**). The line was maintained as a co-isogenic strain on this C57BL/6JCrl background. Heterozygous mice were intercrossed to produce viable and fertile homozygotes (*Mab21l1$^{R51L/R51L}$*). The ratios of offspring were consistent with Mendelian genetics (**S3 Table in S7 File**). *Mab21l1$^{R51L/+}$* heterozygote mice showed anomalous, excavated optic discs (**Fig 4A and 4B**) in 13/13 heterozygotes, confirmed as bilateral in 9/13 (all fundus images are shown in **S6 Fig**). Quantitative analysis of fundal images show the discs were enlarged compared to wild type (n = 8 *Mab21l1$^{R51L/+,}$* n = 4 WT, p = 0.00004,). The excavated optic disc anomaly was observed on Optical Coherence Tomography and histological sectioning (**Fig 4C and 4D**). Intraocular pressure tested in a subset of mice (n = 2 WT, n = 5 *Mab21l1$^{R51L/+}$*, n = 3 *Mab21l1$^{R51L/R51L}$*) was within the normal range, consistent with the optic nerve phenotype being a developmental defect rather than glaucomatous phenomenon. Slit lamp examination of the iris and anterior segment appeared normal and the mice displayed no other apparent abnormalities. All homozygous *Mab21l1$^{R51L/R51L}$* mice had a severe bilateral panocular eye malformation (**Fig 4A and 4B**). These included microphthalmia with disorganised anterior and posterior segments. There was marked hyperplasia of pigmented uveal tissue that obscured any possibility of a fundal view on examination. Histological sectioning revealed abnormalities of the cornea, iris, ciliary body, lens, retina and optic nerve (**Fig 4C**). The most severely affected eyes had only a rudimentary lens and retina, and optic nerve aplasia.

## Discussion

Classical aniridia is a highly distinctive autosomal dominant disorder diagnosed in infancy by the combination of absence of the iris and foveal hypoplasia [32,33]. In adult life a progressive opacification of the cornea results in the relentless loss of their vision; this is currently untreatable and represented a particularly challenging aspect of the disorder for both affected

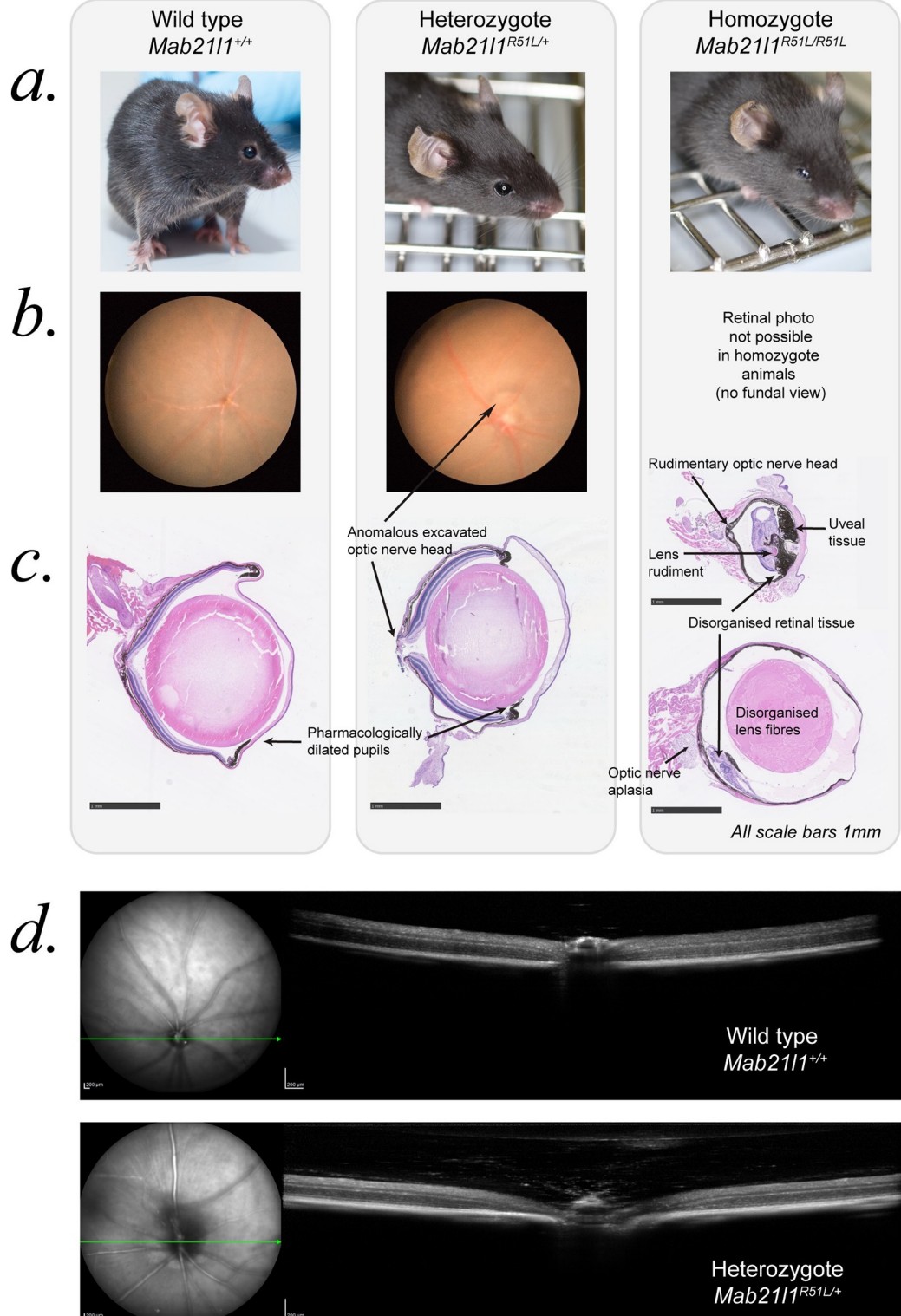

**Fig 4. Mab21l1$^{R51L}$ mouse phenotype.** Left hand column shows wild type (WT) mice, middle column Mab21l1$^{R51L/+}$ heterozygous mice and right hand column homozygous Mab21l1$^{R51L/R51L}$ mice. All mice were examined as young adults, at 2–3 months of age. **(A)** External photographs, showing normal external appearance of heterozygous and severe microphthalmia in homozygous mice. **(B)** Representative retinal photographs showing the anomalous, excavated optic disc phenotype (arrow) present in all the 13 heterozygous mice examined, and none of the WT (further images in **S6 Fig**).

No retinal view (or OCT) was possible in homozygous animals as all eyes were microphthalmic, often with uveal tissue obscuring the cornea. **(C)** H&E-stained wax sections of the mouse eyes, with the abnormal features for each genotype labelled. The excavated optic nerve anomaly of the heterozygous mice is clearly seen. Note these mice had recently-administered dilating drops for fundal examination, but the iris appeared normal on slit lamp examination. Homozygous mice had a severe panocular eye malformation including microphthalmia, severely disorganised retina and uveal tissue, along with hypoplasia or aplasia of both the optic nerve and lens. Eyes from two different age-matched homozygous mice are shown to illustrate the spectrum of microphthalmia. Scale bars = 1 mm. **(D)** Optical coherence tomography (OCT) of WT and heterozygous mice, showing the enlarged, excavated optic nerve, with some persistent fetal vasculature seen above the optic nerve head.

individuals and ophthalmologists. More than 90% of individuals with aniridia have heterozygous mutations detectable at the *PAX6* locus that appear to result in loss-of-function [34–36]. Rare monoallelic missense variants at specific residues within the PAIRED domain of *PAX6* cause a significantly more severe form of classical aniridia with microphthalmia [37] with Fig 1 of that paper demonstrating the striking similarity to the individuals with *MAB21L1* mutations reported here. The phenotypic similarity includes the nature of the iris phenotype (with a spectrum including moderate and profound absence of iris, with the loss not limited to one specific part of the iris such as in Gillespie syndrome), the severity of microphthalmia and the other associated ocular features including cataract, lens instability and foveal hypoplasia. The molecular basis of this worse-than-null phenotypic effect in the PAX6 missense cases is unknown but is assumed to be the consequence of altered PAX6 interaction with DNA and/or co-binding partners such as SOX2 [38–40].

Given the phenotypic similarity of monoallelic missense variants resulting in *PAX6* and Arg51 MAB21L1 substitutions we hypothesize that the developmental function of the wild-type PAX6 and MAB21L1 proteins are interdependent. In this regard, it is interesting to consider similarity between the male-specific sensory rays mis-specification that characterizes the mab-21 mutant class in *C. elegans* [1] and that seen in mab-18 mutants caused by a mutation at the *vab-3* (*PAX6*) locus [41]. It is also striking that four of the five most abundant proteins recovered by immunoprecipitating MAB21L1 were the transcription factors MEIS1, MEIS2, PBX1 and PBX3. These transcription factors act as both activators [42–45] and co-binding partners [46,47] of each other and PAX6. Although these interactions are probably relevant to the developmental role of MAB21L1 it is difficult to link them to disease as they were not significantly altered by either of the Arg51 substitutions we studied.

Using IP-MS we could identify only one protein, TBL1XR1 that interacted with wild-type MAB21L1 but not at all with either mutant form. TBL1XR1 mediates proteasomal degradation of NCor corepressor complex [48]. Western blotting suggested that this interaction with mutant protein was reduced rather than completely ablated (**Fig 3F**). Although disruption of such an interaction is a reasonable candidate for perturbing a developmental transcriptional cascade it should be noted that this interaction would be completely ablated in the individuals with homozygous loss-of-function mutations in MAB21L1 but these individuals have significantly milder anterior segment anomalies than the individuals carrying Arg51 heterozygous missense variants. We do not therefore consider this loss of protein-protein interaction to be the likely mechanism of disease in the affected individuals we present here.

Three proteins, GALNT2, MSI2 and LRRFIP1, showed apparent mutation-specific interactions suggesting a possible gain of function effect (**S5 Fig**). GALNT2 is a N-acetyl-d-galactosamine-transferase 2 which localizes to the Golgi which has not previously been implicated in PAX6 function or eye development. Biallelic loss of function mutations in *GALNT2* [MIM 602274] cause a neurodevelopmental disorder of O-linked glycosylation [MIM: 618885] [49]. LRRFIP1 [MIM: 603256] is an RNA binding protein that binds double stranded RNA [50]. LRRFIP1 has roles both as a viral sensor in innate immunity [51] and as a regulator of

canonical WNT signalling in development [50,52]. There is no direct evidence that LRRFIP1 is involved in eye development or PAX6 function.

MSI/Musashi-2 [MIM 607897] is also an RNA binding protein that regulates the translation of gene products through binding their 3'UTR regions. Its role in both cancer [53] and developmental systems [54–56] has been widely studied and it has been shown to form a complex with SOX2 [57]. This seemed a good candidate as a gain-of-function interaction but we could identify only limited evidence for this using reciprocal IP-MS and the interaction was not detectable using western blot analysis following IP. Musashi-1 was also identified as a MAB21L1 interactor but did not show any difference between mutant and wild-type proteins (S5 Fig). Musashi-1 and -2 are required for normal photoreceptor development [58].

The identification of RNA binding proteins as an overrepresented class in the list of mutation agnostic MAB21L1 interactors may be of significant functional relevance. The crystal structure of MAB21L1 suggested that activation of the nucleotidyltransferase activity required a conformational change similar to that of the mab-21 paralog cGAS (S7 Fig). The authors could demonstrate that MAB21L1 bound double stranded RNA but with significantly lower affinity than cGAS [8]. Our work would support their conclusion that any generic oligonucleotides are unlikely to function as MAB21L1 inducers. They go on to suggest that specific mRNA-RNA-binding protein complexes species may bind to MAB21L1 to induce the enzymatic activity. The fact the LRRFIP1 has cGAS like functions in sensing viral dsRNA in the cytoplasm is interesting but given that this interaction is only seen with Arg51 substitutions make this an unlikely endogenous activator. Our favoured hypothesis is that RNA-bound Musashi-2 functions as the *in vivo* activator, and a competitive antagonistic effect of Musashi-2 binding in the mutant is an important gain-of-function interaction. All the above protein-protein interaction experiments must be treated with caution given that they were performed using GFP-tagged peptides that were very highly and inducibly expressed in HEK293 cells. We suggest that future work should focus on identifying wildtype and mutant MAB21L1-specific interactions under more physiologically and developmentally relevant tissues to identify the molecular basis of the disorder.

There are several notable features regarding the phenotypes associated with monoallelic and biallelic mutation of *Mab21l1* in mice. The phenotype in *Mab21l1*^R51L/+ mice is milder than in humans, but the optic disc anomaly is both seen in human cases and consistent with *PAX6*-associated disease [59]. In the process of this work Seese and colleagues [14] reported the identification of c.152G>T p.(Arg51Leu) variant in MAB21L1 in two affected members of a family which co-segregated with microphthalmia and aniridia. This family appear to be phenotypically very similar to those we have identified and this is further support for the causative nature of substitutions affecting MAB21L1 Arg51. It is interesting that the severe eye malformations in *Mab21l1*^R51L/R51L animals resemble those reported in *Mab21l1* null animals [6]. In contradistinction the eye phenotype in null humans is significantly milder than that seen in mice or indeed heterozygous Arg51 substitutions in humans. Together this suggests that there may be significant differences in *MAB21L1/Mab21l1* dosage sensitivity and disease mechanism between mice and humans.

## Supporting information

**S1 Fig. *Mab21l1* CRIPSR design. (*A*)** Schematic to illustrate the CRISPR-Cas9 sgRNA guide sequences and their relative locations to the Arginine 51 encoding region of the *Mab21l1* locus.(***B***) Sanger sequencing chromatogram of PCR performed using genomic DNA prepared from a gene edited mouse. The *Mab21l1* p.Arg51Leu mutation was introduced (highlighted region), along with the silent substitutions in the flanking regions (red asterisks), which were

specific to the repair template.
(DOCX)

**S2 Fig. Sequences of highly specific *MAB21L1* heterozygous variants associated with microphthalmia and/or aniridia.** An allelic series of *MAB21L1* heterozygous variants at position c.152G was identified in a total of five probands: two familial cases with c.152G>A (p. (Arg51Gln), chromatograms in orange shaded box), two sporadic cases with *de novo* inheritance of either the recurrent variant c.152G>T (p.(Arg51Leu), upper chromatogram in green shaded box) or the novel variant c.152G>C (p.(Arg51Pro), chromatogram in yellow shaded box), and one familial case with unknown genotypic inheritance of the recurrent variant c.152G>T (p.(Arg51Leu), lower chromatogram in green shaded box). Additionally, a sporadic case with unknown genotypic inheritance was heterozygous for the novel variant c.155T>G (p.(Phe52Cys), chromatogram in pink shaded box) in the adjacent 3' codon. The chromatogram for each proband is shown, with the Family ID and pedigree case ID detailed to the right. Sanger sequencing was used to screen for and/or validate the variant in each proband, and to test all of the available relatives (data not shown), which established segregation with the phenotype. The schematic (upper right) illustrates the highly specific positioning of the four variants identified. Nucleotide and amino acid numbering is based on GenBank: NM_005584.5 and GenPept: NP_005575.1, respectively.
(DOCX)

**S3 Fig. Dominant and recessive variants of MAB21L1 and MAB21L2.** Schematic representations of the linear form of MAB21L1 (blue filled bar) and MAB21L2 (purple filled bar) are shown, with the first and final amino acids numbered for each protein. For both MAB21L1 and MAB21L2 the linear positions of all published pathogenic variants are detailed on each cognate protein schematic, with the dominant heterozygous variants shown above and the recessive biallelic variants shown below. The MAB21L1 variants identified in this study are all dominantly inherited and are shown in red text. Abbreviations: *dn*, de novo. Nucleotide and amino acid numbering are based on GenBank NM_005584.5 and GenPept NP_005575.1, respectively.
(DOCX)

**S4 Fig. Reciprocal IP using TBL1XR1 antibody.** Western blot analysis of the anti-TBL1XR1 IP using the anti-GFP antibody was unable to detect interaction with wild-type or mutant forms of MAB21L1.TBL1XR1 was detected in all the pull down used as control for pull down experiment.
(DOCX)

**S5 Fig. Mutant and WT-specific protein-protein interactions from IP-MS.** A. Graphs of the log-transformed quantitative mass spectroscopy results of biological triplicates of immunoprecipitates of control (GFP), tagged wild-type (WT) and mutant MAB21L1 (R51L and R51Q) from HEK293 cells which identified as single wild-type specific interactor (TBL1XR1) which is a component of the NCor complex. Two other subunits of the NCor complex (NCOR & HDAC3) are shown for comparison. B. Graphs of the three mutant specific interactors (GALNT2, LRRFIP1, MSI2/Musashi-2) and /Musashi-1, a close homolog of MSI2, which shows interaction with all forms of MAB21L1.
(DOCX)

**S6 Fig. Fundus images showing the optic nerve anomaly of Mab21l1 R51L heterozygous mice.** Retinal photographs of additional Mab21l1 R51L heterozygous and wild type mice, supplementing the representative, annotated images shown in Fig 4. (A) Photographs from 8

heterozygous Mab21l1 R51L/+ mice, left and right eyes, showing enlarged, excavated anomalous optic nerve heads, often with prominent persistent fetal vasculature. (B) Photographs from 5 wild type mice from the same colony (littermate or cousin controls), showing normal optic nerve heads, illustrating the range of normal. (C, below) On each line, a cartoon of the first fundus image is shown, illustrating the optic disc (black circle) and major blood vessels (radial lines), followed by fundus photographs from one eye of 3 different mice. (i) Mab21l1R51L/+ heterozygous mice (images from 3 mice representative of the optic nerve phenotype taken from images above in (A), versus (ii) age-matched wild type littermate/cousin controls.
(DOCX)

**S7 Fig. Human mab-21 paralogs: Peptide sequence and genomic features.** A. Phylogenetic tree of the 11 human mab-21 paralogs and protein alignment (B) and genomic organisation (C) of MAB21L1, MAB21L2 and mab-21. The alignment and phylogenetic tree were generated using MUSCLE https://www.ebi.ac.uk/Tools/msa/muscle/
(DOCX)

**S1 File. Supplemental clinical descriptions.**
(DOCX)

**S2 File. Supplemental materials and methods.**
(DOCX)

**S3 File. Supplementary raw data table: Pyrophosphate assay.**
(XLSX)

**S4 File. Supplementary raw data table: Mass spectrometry.**
(XLSX)

**S5 File.**
(DOCX)

**S6 File.**
(DOCX)

**S7 File.  S1 Table. Oligonucleotides used in the *MAB21L1*/*Mab21l1* study: Sequence and protocol details.** Underlined sequence denotes universal tags with no homology to *MAB21L1*. Further details of the biological relatedness microsatellite PCR protocol are available at http://www.faa.gov/data_research/research/med_humanfacs/oamtechreports/2000s/media/200614.pdf. **S2 Table**. MAB21L1 variant nomenclature validation. (https://variantvalidator.org/). **S3 Table. Mendelian ratios.** Comparison of the observed versus expected ratios of genotypes from intercrosses of the *Mab21l1* R51L line mice (n = 14 litters of each type), establishing that the observed ratios were consistent with Mendelian genetics. **S4 Table. FoldX values.** Molecular modelling performed using FoldX (Delgado et al., 2019) in order to assess the impact of MAB21L1 and MAB21L2 substitutions on protein stability. Nearly all the mutations are destabilizing to protein structure.
(DOCX)

**S1 Raw images.**
(TIF)

## Acknowledgments

We thank the patients and families for their participation. For rodent expertise we thank Lorraine Rose, Anna Thornburn, John Campbell, Jacek Mendrychowski and the staff of Central Bioresearch Services.

## Author Contributions

**Conceptualization:** Veronica Van Heyningen, David R. FitzPatrick.

**Data curation:** Hemant Bengani, Morad Ansari, Joseph A. Marsh.

**Formal analysis:** Hildegard Nikki Hall, Hemant Bengani, Joseph A. Marsh, Graeme R. Grimes.

**Funding acquisition:** David R. FitzPatrick.

**Investigation:** Hemant Bengani, Brian P. Brooks.

**Methodology:** Hemant Bengani, Alex von Kriegsheim, Lisa McKie, Joe Rainger.

**Project administration:** David R. FitzPatrick.

**Resources:** Robert B. Hufnagel, Giuseppe Damante, David Moore, Jamalia Rahmat, Catia Mio, Moira Blyth, Wee Teik Keng, Lily Islam, Meriel McEntargart, Marcel M. Mannens, Brian P. Brooks.

**Supervision:** David R. FitzPatrick.

**Validation:** Hemant Bengani, Joe Rainger.

**Writing – original draft:** David R. FitzPatrick.

**Writing – review & editing:** Hildegard Nikki Hall, Hemant Bengani, Joseph A. Marsh, Joe Rainger.

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
