## [Decision Letter · Decision Letter 0]

31 May 2022

PONE-D-22-11688Monoallelic Variants Resulting in Substitutions of MAB21L1 Arg51 Cause Aniridia and MicrophthalmiaPLOS ONE

Dear Dr. FitzPatrick,

Thank you for submitting your manuscript to PLOS ONE. After careful consideration, we feel that it has merit but does not fully meet PLOS ONE’s publication criteria as it currently stands. Therefore, we invite you to submit a revised version of the manuscript that addresses the points raised during the review process.

We look forward to receiving your revised manuscript.

Kind regards,

Anand Swaroop

Academic Editor

PLOS ONE

Journal Requirements:

Additional Editor Comments:

This is an excellent manuscript, which can be improved further by following the suggestions of the reviewers, especially Reviewer 1. Senior authors are experts in the area and should expand a little more in Introduction and Discussion to make the manuscript even more exciting and significant.

Reviewers' comments:

Reviewer's Responses to Questions

**Comments to the Author**

1. Is the manuscript technically sound, and do the data support the conclusions?

Reviewer #1: Yes

Reviewer #2: Yes

2. Has the statistical analysis been performed appropriately and rigorously? 

Reviewer #1: N/A

Reviewer #2: Yes

3. Have the authors made all data underlying the findings in their manuscript fully available?

Reviewer #1: Yes

Reviewer #2: Yes

4. Is the manuscript presented in an intelligible fashion and written in standard English?

Reviewer #1: Yes

Reviewer #2: Yes

5. Review Comments to the Author

Reviewer #1: This manuscript reports monoallelic variants in a single MAB21L1 residue cause a specific eye phenotype. While a useful report, several simple changes would enhance.

1. The authors should distill the literature on Mab-21 genes, and specifically MAB21L1, into a few pithy paragraphs, to fully introduce the topic. Description of broader MAB21L1 mutation phenotypes, the diverse animal models and their phenotypes, together with published data from other groups, would place their own findings in better context. With such a background, the reader would wish to know if the current patient cohort exhibited aphakia, scrotal anomalies, etc.

2. A strength of this manuscript is that the authors created a knock-in for Mab21l1, which represents a very considerable amount of work. However, description of this and analysis of the mutant, is underdeveloped, missing an opportunity to impress the reader/reviewer regarding the range of approaches used.

3. Use of the term ‘aniridia’ (title, and elsewhere) represents quite loaded phrasing. Depending on the phenotype present, this may be misleading, particularly as it implies a biological link to PAX6 although no such data are provided. Indeed, previous studies provide zero evidence for Pax6 involvement (see below).

As the authors know well, virtually all reported aniridia cases are attributable to PAX6. The odd exception, generally in reports with an n of 1, claim the same phenotype on the basis of low quality clinical images.

If absence of iris tissue represents “aniridia”, it would be much better to include high quality images demonstrating the PAX6 phenotypic spectrum (including retro-illumination ‘shots’ illustrating the lens), side by side with comparable and much higher quality images from their cohort (even from a single pedigree). Presentation of such data would support accurate phenocopying of loss of PAX6. Alternatively, they could explain they see loss of iris tissue, possibly profound loss, and discuss whether this does or does not resemble PAX6-induced disease.

4. The above is important, due to elegant experiments (Development, 2003) which demonstrated that Mab21l1 murine homozygotes fail to develop lens placodes, and consequently exhibit complete loss of the iris. So heterozygous patients having a partial loss of iris phenotype would be consistent. However, Yamada et al. also demonstrated that Pax6 expression was completely preserved (Foxe3 was profoundly altered), suggesting that this is not “classical aniridia” but a loss of iris phenotype. These elements should surely be discussed/explored in the current manuscript. Indeed, switching emphasis from Pax6, clinically and as a mechanistic candidate, provides opportunities to discuss alternatives for which a strong biological basis exists.

5. For instance, another publication identified genes dysregulated in zebrafish mab21l1 mutants. These included both Spalt and Sox transcription factors, and like the prior Foxe3 data, the implications should be discussed.

6. The authors used HEK cells to characterize MAB21L1 mutation, observing few differences in expression compared to wildtype. It would be helpful to comment on this finding in the context that renal phenotypes are not a feature of Mab21l1 mutation, and renal expression of MAB21L1 is very low (GTExPortal).

7. The penultimate sentence of the Figure 1 legend incorrectly states that the upper and lower photos show progression of keratopathy. The photos in fact illustrate phthisis - a much more profound phenotype in which the eye progressively shrinks in size. When phthisis occurs, multiple other phenotypes are induced including frequent corneal vascularization and opacification.

Reviewer #2: The manuscript by Hildegard Nikki Hall et al. presents a comprehensive and accurate analysis of novel findings related to MAB21L1 missense variants as the cause of Aniridia and Microphthalmia. The manuscript is well-written, contains all needed information either in the manuscript body or in the supplementary information section, and includes a detailed discussion on the obtained results. I have only a minor comment regarding Figure 1:

- In panel "a", there is a symbol for "developmental delay" which is not found in any of the pedigrees. Please check.

- There are many marks in each symbol, including the genotype as a circle in the middle of the symbol. That can cause confusion. I would suggest writing the genotype beneath each symbol of a recruited individual.

6. PLOS authors have the option to publish the peer review history of their article (what does this mean?). If published, this will include your full peer review and any attached files.

Reviewer #1: No

Reviewer #2: No

---

## [Author Response · Author response to Decision Letter 0]

6 Sep 2022

PONE-D-22-11688 Monoallelic Variants Resulting in Substitutions of MAB21L1 Arg51 Cause Aniridia and Microphthalmia 

Reviewer's Responses to Questions 

Comments to the Author 

Reviewer #1: This manuscript reports monoallelic variants in a single MAB21L1 residue cause a specific eye phenotype. While a useful report, several simple changes would enhance. 

1. 

The authors should distil the literature on Mab-21 genes, and specifically MAB21L1, into a few pithy paragraphs, to fully introduce the topic. 

Description of broader MAB21L1 mutation phenotypes, the diverse animal models and their phenotypes, together with published data from other groups, would place their own findings in better context. 

The C elegans and murine phenotypes associated with loss of function are already mentioned in the introduction. We have expanded the mouse description to now read: 

“Mab21l1 null mice are viable but show severe bilateral microphthalmia with a small malformed lens and absence of the iris and ciliary body [PMID 12642482]. Null mice also show delayed calvarial development and male infertility with hypoplasia of the preputial glands” 

We have added the following sentences about the recently reported zebrafish model: 

“Homozygosity for an early frameshift mutation in zebrafish mab21l1 resulted in a late embryonic degeneration of the cornea and subsequently the lens.” 

With such a background, the reader would wish to know if the current patient cohort exhibited aphakia, scrotal anomalies, etc 

As far as we are aware there is no evidence of aphakia or genital anomalies in the individuals reported in our manuscript. A sentence to this effect has been added to the clinical section of the results. 

2. 

A strength of this manuscript is that the authors created a knock-in for Mab21l1, which represents a very considerable amount of work. However, description of this and analysis of the mutant, is underdeveloped, missing an opportunity to impress the reader/reviewer regarding the range of approaches used. 

We agree that the mouse work could be developed further. Whilst a range of approaches were indeed used, not all were fruitful and we have only included the useful and relevant findings here. We have expanded the Supplemental Fig S6 to expand the figure concerning the mouse optic disc phenotype. 

3. 

Use of the term ‘aniridia’ (title, and elsewhere) represents quite loaded phrasing. Depending on the phenotype present, this may be misleading, particularly as it implies a biological link to PAX6 although no such data are provided. Indeed, previous studies provide zero evidence for Pax6 involvement (see below). 

As the authors know well, virtually all reported aniridia cases are attributable to PAX6. The odd exception, generally in reports with an n of 1, claim the same phenotype on the basis of low quality clinical images. 

If absence of iris tissue represents “aniridia”, it would be much better to include high quality images demonstrating the PAX6 phenotypic spectrum (including retro-illumination ‘shots’ illustrating the lens), side by side with comparable and much higher quality images from their cohort (even from a single pedigree). Presentation of such data would support accurate phenocopying of loss of PAX6. Alternatively, they could explain they see loss of iris tissue, possibly profound loss, and discuss whether this does or does not resemble PAX6-induced disease. 

We agree with the above comment, and we also consider classical aniridia to be a PAX6-associated disorder. For this reason we have been careful not to use the term classical aniridia in association with the cases reported here. However, it is important to appreciate that all but one of these cases were referred to our study by very experienced clinicians over many years with a primary diagnosis of aniridia and with a clinical suspicion that there may be a cryptic PAX6 mutation. We regret that we have not been able to obtain better quality clinical images of some of the pedigrees. We have however included, in an updated Figure 1, a new clinical image of one of the individuals which better shows complete loss of iris tissue via retroillumination (family 511), and enlarged some of the iris photos showing partial loss (family 1434). We have added the qualifiers of “profound” and “partial” aniridia to the descriptions in the first section of the Results, and included a section summarising the clinical phenotype features outlined in Table 1. 

The overall phenotype of these cases (including iris phenotype, associated significant microphthalmia, and other associated features) are strikingly similar to the severe PAX6 missense cases reported in our 2019 paper PMID 31700164 (Figure 1) and this has been emphasised in the first paragraph of the Discussion. 

The phenotypic similarity includes not just the nature of the iris loss phenotype (with a spectrum including moderate and profound absence of iris, with the loss not limited to one specific part of the iris such as in Gillespie syndrome), but also the microphthalmia and other associated ocular features including cataract, lens instability and foveal hypoplasia. 

4. 

The above is important, due to elegant experiments (Development, 2003) which demonstrated that Mab21l1 murine homozygotes fail to develop lens placodes, and consequently exhibit complete loss of the iris. So heterozygous patients having a partial loss of iris phenotype would be consistent. However, Yamada et al. also demonstrated that Pax6 expression was completely preserved (Foxe3 was profoundly altered), suggesting that this is not “classical aniridia” but a loss of iris phenotype. These elements should surely be discussed/explored in the current manuscript. Indeed, switching emphasis from Pax6, clinically and as a mechanistic candidate, provides opportunities to discuss alternatives for which a strong biological basis exists. 

PAX6 is rightly known as a master regulator of eye development. The reviewer will be very aware that the molecular effectors of many developmental functions of PAX6 at different stages of eye development are far from clear. In essence, we are suggesting that MAB21L1 may be one of these effectors. Our hypothesis is that the Arg51 substitutions disrupt this effector function via an effect that is different to that associated with loss of MAB21L1 function. This is completely compatible with PAX6 being “upstream” of MAB21L1. 

5. 

For instance, another publication identified genes dysregulated in zebrafish mab21l1 mutants. These included both Spalt and Sox transcription factors, and like the prior Foxe3 data, the implications should be discussed. 

The zebrafish data is indeed very interesting however our paper is focused on the very striking human genetic features of this disorder and we made a decision when writing the paper that is was not appropriate for us to speculate too much regarding the precise molecular basis of the effect. We respectfully suggest this would be better done by other investigators who will, we hope, follow up on our work. 

6. 

The authors used HEK cells to characterize MAB21L1 mutation, observing few differences in expression compared to wildtype. It would be helpful to comment on this finding in the context that renal phenotypes are not a feature of Mab21l1 mutation, and renal expression of MAB21L1 is very low (GTExPortal). 

Although HEK293 cells were derived from human embryonic kidney tissue, they have been considered cells of complex phenotype for many years. They express markers of neuronal, renal and adrenal progenitors with almost no significant characteristics of adult kidney cells (see PMID: 2602690). We completely accept their limitations, but their tolerance of high-levels of exogenous gene expression is extremely useful experimentally. In our functional studies of genes involved in human eye malformations we also find it helpful that HEK293 cell endogenously express PAX6 and SOX2 at detectable levels, possibly, as part of their neuronal phenotype. We have added the following sentence to the discussion: 

“All of the above protein-protein interaction experiments must be treated with caution given that they were performed using GFP-tagged peptides that were very highly and inducibly expressed in HEK293 cells. We suggest that future work should focus on identifying wildtype and mutant MAB21L1-specific interactions under more physiologically and developmentally relevant tissues in order to identify the molecular basis of the disorder.” 

7. 

The penultimate sentence of the Figure 1 legend incorrectly states that the upper and lower photos show progression of keratopathy. The photos in fact illustrate phthisis - a much more profound phenotype in which the eye progressively shrinks in size. When phthisis occurs, multiple other phenotypes are induced including frequent corneal vascularization and opacification. 

In the Clinical Descriptions the phthisis of the right eye is described, and we have amended the Figure 1 legend. The examining clinician describes the right eye as phthisical and the left eye as demonstrating progressive pannus, corneal oedema and band keratopathy. We are therefore reluctant to label both eyes as phthisical given this report but have removed the mention of progressive keratopathy as we agree it could potentially be misleading. 

Reviewer #2:  

The manuscript by Hildegard Nikki Hall et al. presents a comprehensive and accurate analysis of novel findings related to MAB21L1 missense variants as the cause of Aniridia and Microphthalmia. The manuscript is well-written, contains all needed information either in the manuscript body or in the supplementary information section, and includes a detailed discussion on the obtained results. I have only a minor comment regarding Figure 1: 

- In panel "a", there is a symbol for "developmental delay" which is not found in any of the pedigrees. Please check. 

- There are many marks in each symbol, including the genotype as a circle in the middle of the symbol. That can cause confusion. I would suggest writing the genotype beneath each symbol of a recruited individual. 

We broadly agree with these comments and have amended the Figure 1. The number of pedigree symbols has been reduced, removing developmental delay and oculocutaneous albinism. The annotation for the genotypes has been simplified for clarity, removing the confusing “N” in the previous version.

---

## [Decision Letter · Decision Letter 1]

7 Oct 2022

Monoallelic Variants Resulting in Substitutions of MAB21L1 Arg51 Cause Aniridia and Microphthalmia

PONE-D-22-11688R1

Dear Dr. FitzPatrick,

We’re pleased to inform you that your manuscript has been judged scientifically suitable for publication and will be formally accepted for publication once it meets all outstanding technical requirements.

Kind regards,

Anand Swaroop

Academic Editor

PLOS ONE

Additional Editor Comments (optional):

Reviewers' comments:

Reviewer's Responses to Questions

**Comments to the Author**

1. If the authors have adequately addressed your comments raised in a previous round of review and you feel that this manuscript is now acceptable for publication, you may indicate that here to bypass the “Comments to the Author” section, enter your conflict of interest statement in the “Confidential to Editor” section, and submit your "Accept" recommendation.

Reviewer #1: All comments have been addressed

Reviewer #2: All comments have been addressed

2. Is the manuscript technically sound, and do the data support the conclusions?

Reviewer #1: Partly

Reviewer #2: Yes

3. Has the statistical analysis been performed appropriately and rigorously? 

Reviewer #1: N/A

Reviewer #2: Yes

4. Have the authors made all data underlying the findings in their manuscript fully available?

Reviewer #1: Yes

Reviewer #2: Yes

5. Is the manuscript presented in an intelligible fashion and written in standard English?

Reviewer #1: Yes

Reviewer #2: Yes

6. Review Comments to the Author

Reviewer #1: (No Response)

Reviewer #2: None.

7. PLOS authors have the option to publish the peer review history of their article (what does this mean?). If published, this will include your full peer review and any attached files.

Reviewer #1: No

Reviewer #2: No

---

## [Editor Report · Acceptance letter]

4 Nov 2022

PONE-D-22-11688R1 

Monoallelic Variants Resulting in Substitutions of MAB21L1 Arg51 Cause Aniridia and Microphthalmia 

Dear Dr. FitzPatrick:

I'm pleased to inform you that your manuscript has been deemed suitable for publication in PLOS ONE. Congratulations! Your manuscript is now with our production department. 

Kind regards, 

on behalf of

Dr. Anand Swaroop 

Academic Editor

PLOS ONE